# Novel Carboxylated Chitosan-Based Triptolide Conjugate for the Treatment of Rheumatoid Arthritis

**DOI:** 10.3390/pharmaceutics12030202

**Published:** 2020-02-26

**Authors:** Lan Zhang, Min Yan, Kun Chen, Qikang Tian, Junying Song, Zijuan Zhang, Zhishen Xie, Yong Yuan, Yaquan Jia, Xin Zhu, Zhenqiang Zhang, Xiangxiang Wu, Huahui Zeng

**Affiliations:** 1Academy of Chinese Medicine Sciences, Henan University of Chinese Medicine, Zhengzhou 450046, China; 2Pharmacy College, Henan University of Chinese Medicine, Zhengzhou 450046, China; 3School of Basic Medicine, Henan University of Chinese Medicine, Zhengzhou 450046, China

**Keywords:** triptolide, toxicity, water solubility, carboxylmethyl chitosan, drug carrier system

## Abstract

A new platform for triptolide (TP) delivery was prepared by conjugating TP to a carboxylmethyl chitosan (CMCS). Compared with the natural TP, the TP-conjugate (TP-CMCS) containing TP of ~5 wt% exhibited excellent aqueous solubility (>5 mg/mL). Results of in vitro experiments showed that TP-CMCS could relieve TP-induced inhibition on RAW264.7 cells and apoptosis, respectively. Compared with the TP group, TP-CMCS could effectively alleviate the toxicity injury of TP and decreased the mortality rate of the mice (*p* < 0.05). TP-CMCS did not cause much damage to the liver (AST and ALT) and kidney (BUN and CRE) (*p* < 0.05). After administration, the levels of IL-6, IL-1β, and TNF-α decreased, and the arthritis detumescence percentages increased significantly, and the bony erosion degree was distinctly decreased in the TP-CMCS groups and TP group. Our results suggested that TP-CMCS was a useful carrier for the treatment of RA, which enhanced aqueous solubility of free TP and reduced drug toxicity in vitro and in vivo.

## 1. Introduction

Rheumatoid arthritis (RA) is an autoimmune inflammatory disease characterized by erosion of the cartilage and bone, by synovial hyperplasia, pain and swelling, and even joint deformity [1]. The hyperproliferation and infiltration of synovial cells are caused by the proinflammatory cytokines and other inflammatory mediators, such as TNF-a, IL-1β, IL-6, which are secreted from macrophages, T cells, and B cells [2]. Moreover, the cartilage and bone damage is caused by osteoclasts, activated synovial fibroblasts and chondrocytes in inflammatory arthritis [3]. Currently available anti-rheumatic drugs, including non-steroidal drugs, glucocorticoids, and biological agents, cannot cure RA radically but only relieve its symptoms and progression [4,5]. In China, herbal medicines have been extensively utilized as alternative RA drugs for decades [6,7]. However, the traditional herbs contain a lot of ingredients which may result in serious side-effects [8]. Therefore, the functions of every ingredient must be further defined to improve the therapeutic performance of the herbs.

Triptolide (TP), extracted from Tripterygium wilfordii Hook F (TWHF), has been widely utilized to treat inflammatory and autoimmune diseases, such as RA and systemic lupus [9,10]. Generally, it is more effective in treatment of RA than in that of systemic lupus. TP can prevent bone damage through the RANKL-mediated ERK/Akt pathway and induce T-cell apoptosis in collagen induced arthritis (CIA) mice [11,12,13]. However, its use is greatly restricted by poor aqueous solubility (0.017 mg/mL), rapid in vivo efflux, and various toxic effects (LD50, 0.8 mg/kg), which are mainly associated with damaging the liver, spleen, kidneys, and blood circulatory system [10]. Thus, it is desirable to explore an effective and safe drug carrier to improve the pharmacological performance of triptolide.

To address these issues, an immunosuppressant conjugate should be developed for RA treatment. Currently, chitosan has been widely explored as a drug delivery carrier. However, its applications are limited by its poor solubility in water. Carboxylmethyl chitosan (CMCS), a chemically modified version of chitosan, was used as drug delivery carrier to form 1,4-succinate triptolide-carboxylated chitosan conjugate (TP-CMCS), owing to its odegradability, biocompatibility, low toxicity, as well as reasonable cost [14,15,16]. CMCS is a water-soluble polysaccharide derived by partial N-carboxymethylation of chitosan. Furthermore, the weakly negatively charged chains have avoided undesirable cytotoxicity and prevented uptake of TP from the macrophage and reticuloendothelial system. Here, the novel formulation of triptolide was developed to reduce its cytotoxicity and guarantee its effect for RA treatment, which was confirmed by the toxicity assay in vitro/vivo, the clinical indexes of the anti-inflammatory effect, and the histological analyses of synovial joints.

## 2. Materials and Methods

### 2.1. Materials and Animals

Triptolide was purchased from Xi’an Haoxuan Biotechnology Co. Ltd. (Xi’an, China). CMCS was obtained from Aladdin (Beijing, China) and was further purified using dialysis. All other reagents and solvents were purchased from commercial suppliers with analytical grade. The murine macrophage RAW264.7 cells (Manassas, VA, USA) were incubated in DMEM (Dulbecco’s Modified Eagle Medium) containing 10% fetal bovine serum and 1% penicillin and streptomycin (Gibco, Carlsbad, CA, USA). Cells were incubated at 37 °C with 5% CO_2_ in a humidified atmosphere. Kunming mice (20–22 g) were obtained from the Qinglongshan laboratory animal farm, Nanjing City. The mice were allowed to acclimatize for one week in a light-dark cycle environment at 25 ± 1 °C for 12 h and provided with water and food. All animal studies were performed in accordance with a protocol approved by the Institute’s Animal Care and Use Committee (Approval ID: DWLL16020047; Date: 3 June 2016).

### 2.2. Synthesis of Triptolide Analog (TPS) [17]

To a solution of TP (1.80 g, 5 mmol) in pyridine (10 mL) were added succinic anhydride (2.00 g, 20 mmol) and DMAP (4-dimethylaminopyridine, 0.12 g, 1 mmol). The mixed solution was vigorously stirred at 25 °C for 24 h under nitrogen condition. The reaction was monitored using the TLC (Thin-layer Chromatography) method with chloroform-methanol (20:1, *v*/*v*) and 2% 3,5-dinitrobenzoic acid in ethanol and 8% sodium hydroxide in ethanol. After removal of most of the solvent under reduced pressure, the residue was added with CH2Cl2 and then washed with saturated copper sulfate and water, respectively. The dichloromethane layer was dried with anhydrous sodium sulfate and then filtered. After removal of dichloromethane, the residue was loaded on silica gel column chromatography for further purification (DMC/MeOH, 20:1–15:1) to give TPS (1.80 g, 79.3%) as a white solid.

TPS: ^1^H NMR (500 MHz, DMSO-d6) δ 4.98 (s, ^1^H), 4.76–4.87 (m, *J* = 4.82 Hz, ^2^H), 3.95 (d, *J* = 3.95 Hz, 1H), 3.69 (d, *J* = 3.69 Hz, ^1^H), 3.55 (d, *J* = 3.56 Hz, ^1^H), 2.61–2.63 (m, ^1^H), 2.55–2.58 (m, *J* = 2.56 Hz, 2H), 2.48–2.52 (m, *J* = 2.50 Hz, ^2^H), 2.20–2.25 (m, *J* = 2.21 Hz, ^1^H), 2.10–2.14 (d, *J* = 2.10 Hz, ^1^H), 1.96 (m, ^1^H), 1.85–1.90 (m, *J* = 1.88 Hz, ^1^H), 1.77–1.83 (m, *J* = 1.80 Hz, ^1^H), 1.28–1.32 (m, *J* = 1.30 Hz, ^2^H), 0.92 (s, ^3^H), 0.87 (d, *J* = 0.86 Hz, 3H), 0.75 (d, *J* = 0.74 Hz, ^3^H); ^13^C NMR (125 MHz, DMSO-d6) δ 173.73, 173.57, 172.00, 162.64, 123.58, 71.39, 70.69, 63.75, 63.21, 61.37, 59.71, 55.41, 54.89, 35.50, 29.61, 29.34, 29.15, 27.55, 22.80, 17.87, 17.05, 16.97, 14.27; ESI-MS *m*/*z* calcd for C24H28O9 [M^+^] 460.1720, found 460.1716.

### 2.3. Preparation of TP-CMCS Conjugates

TPS (100 mg, 0.22 mmol), NHS (100 mg, 4 eq; *N*-hydroxy succinimide) and EDC (90 mg, 2 eq; 1-ethyl-3-(3-dimethylaminopropyl) carbodiimide hydrochloride) were dissolved in anhydrous DMSO (9 mL), and stirred for 4 h at room temperature to afford TPS-NHS (an NHS ester). Then, the TPS-NHS solution was added dropwise to a CMCS (700 mg, 40 kDa) in a mixture of DMSO (41 mL) and borate buffer (70 mL, pH 10). The reaction solution was stirred for 24 h at room temperature and then was diluted using cold distilled water (d-water, 2×), followed by dialysis against d-water using a dialysis membrane (molecular weight cut off: 1500 Da). TP-CMCS was frozen drying into white solid.

### 2.4. The Weight Percentage of Triptolide in the Conjugate

TP-CMCS (10 mg) was added to 1.0 mL sodium hydroxide solution (6 mol/L) at room temperature for 1 h. The mixture was neutralized with hydrochloric acid (6 mol/L) and then extracted with dichloromethane. After removing solvent, the resulting residue was subsequently analyzed using HPLC (High Performance Liquid Chromatography, C18 column, 4.6 × 250 mm, Agilent; Waldbronn, Germany) with acetonitrile/water (33:67, 1 mL/min). The UV absorption is at a wavelength of 218 nm.

### 2.5. Properties Studies

The solubility of TP-CMCS was evaluated in redistilled water. An excess of TP-CMCS was added to the redistilled water, and vortexed for 5 min, sonicated for 2 min, and centrifuged for 10 min at 14,000 rpm. The supernatant was collected and analyzed as described above.

Drug stability assay was performed in plasma, PBS (Phosphate-buffered Saline, pH 7.4), and cell culture medium (containing 10% FBS) at 37 °C for 20 h. Then, 40 mg of TP-CMCS was dissolved in 10 mL of the above solutions. After this, 500 μL of samples were taken at predetermined times. The released free TP was subsequently extracted by using 1 mL of dichloromethane and analyzed using HPLC.

### 2.6. In Vitro Cytotoxicity Study

RAW264.7 cells were cultured on 96-well plates with 200 μL of DMEM complete medium at 1 × 104 of cells per well. After 24 h, the culture medium was replaced with 200 μL of TP (6.875–888 nmol/L in DMEM complete medium with <1‰ DMSO) or TP-CMCS (at equivalent TP concentration) solutions and cultured for 24 and 48 h, respectively. Cell viability was tested using CCK-8. The plates were incubated for another 1 h. Optical density was measured using a microplate reader (Thermo, Waltham, MA, USA) under 450 nm absorbance values. The cell viability was expressed as below: 
Cell viability (%) = [*A*_(sample)_ − *A*_(blank)_]/[*A*_(control)_ − *A*_(blank)_] × 100

where *A*_(sample)_ and *A*_(control)_ represent the absorbance values for the sample groups and the control groups, respectively. The IC_50_ values of 50% inhibitory concentration of TP and TP-CMCS on RAW264.7 cells incubated for 24 and 48 h were calculated uisng GraphPad Prism7 software.

### 2.7. Analysis of Apoptosis

The cells were cultured with TP (27.5–222 nM) or TP-CMCS (at equivalent TP concentration) for 24 h. After this, the cells were harvested and rinsed with PBS in triplicate. The cells were resuspended in binding buffer (500 μL) containing PI (10 μL) and Annexin V-FITC (5 μL) for 5 minutes. The cell apoptosis rate was determined using Cytoflex (Beckman Coulter, Brea, CA, USA).

### 2.8. In vivo Toxicity Study

In vivo toxicity of TP-CMCS was evaluated using normal Kunming mice. The mice were randomly divided into ten groups (*n* = 8). The mice were administered once every 2 days, via an intravenous injection in the tail, with TP (2 mg/kg, 1 mg/kg, 0.5 mg/kg, 0.25 mg/kg), TP-CMCS (at equivalent to TP concentration), and saline (Control), respectively. The fatality rate and change of body weight were assessed daily for 14 days.

### 2.9. Preparation of Murine CIA Model

The murine CIA model was prepared as follows: Bovine type II collagen (3 mL, 2 mg/mL) was emulsified in complete Freund’s adjuvant (3 mL, 1 mg/mL). The emulsified mixture was injected intradermally on the mouse’s tail. The booster immunization was performed by using incomplete Freund’s adjuvant on the 21st day after the primary immunization. The progression of RA was monitored by scoring the paw in accordance with the published standard [18]. During the preparation and treatment of CIA mice, paws of mice were scored from 0 to 4 for severity of swelling. Briefly, 0 point was no swelling and deformation, and 4 points were the most serious swelling and deformation. The total scores for each mouse were not more than 16 points. All the animal protocols were approved by the Institute’s Animal Care and Use Committee and conformed to the Guide for the Care and Use of Laboratory Animals (Approval ID: DWLL16020047; Date: 3 June 2016).

### 2.10. Therapeutic Effect on CIA Model

The model mice were randomly divided into five groups (*n* = 10) receiving saline, TP-CMCS (0.5 mg/kg), TP-CMCS (1 mg/kg), TP-CMCS (2 mg/kg), and TP (0.5 mg/kg). Each mouse was administered via tail vein injections once every 2 days. The first dose was administered on the 3rd day after grouping. During the treatment, body weight, foot volume, and arthritis deformation scoring of CIA mice were measured every 3 days or 5 days. After the treatment, blood samples were collected from the eyes; arthritic paws were collected for histological examination. The serum was isolated to detect the levels of pro-inflammatory cytokines TNF-α, IL-6, and IL-1β using ELISA (Enzyme-linked immunosorbent assay). The hepatic and renal functions were determined using AST (aspartate aminotransferase), ALT (serum alanine aminotransferase), Crea (creatinine), and BUN (blood urea nitrogen) kits.

### 2.11. Histological Examination

The right tibiae of each group were collected and fixed with 4% formalin and decalcified with 10% EDTA. Tissue sections were prepared by dehydration with gradient alcohol, paraffin embedding, and pathological section. Paraffin slices were examined histologically using an optical microscope after hematoxylin and eosin (HE) staining.

### 2.12. Data Analysis

All data were expressed as means ± SD. Statistical analysis was performed using independent Student’s *t*-test. *p* values were set at less than 0.05, which was considered statistically significant.

## 3. Results and Discussion

### 3.1. Synthesis and Characterization of TP-CMCS

Triptolide was covalently attached to carboxymethyl chitosan via a short linker that is known to be cleaved to release TP in vitro with several hours of half-life [19]. TP-CMCS was synthesized as follows (Scheme 1): (1) esterification reaction between TP and succinic anhydride (the cleavable linker) [17], and (2) conjugating between the esterified TP and CMCS [20]. The conjugate was confirmed using ^1^H-NMR and infrared (IR) spectrum, which showed corresponding peaks to both TP and CMCS (Appendix A). The weight percentage (wt%) of TP in the TP-CMCS conjugate was ~5 wt%. The water solubility of TP-CMCS was determined to be approximately 5 mg/mL. As expected, TP-CMCS has significantly higher water solubility (~15 fold) than the native TP (~0.017 mg/mL), indicating that conjugation of TP to CMCS effectively improved water solubility of TP and enabled elimination of DMSO or Tween 80-based formulation.

To verify whether or not parent TP can be cleaved from the conjugate, the stability of TP-CMCS was determined with incubation in mouse plasma, PBS (pH 7.4), and cell culture medium (10% FBS), respectively (Figure 1). Approximately 4% of TP was fast released from the conjugate within 1 h post incubation in PBS; subsequently, a slow release process occurred during the remaining 49 h, whereas a similar release tendency was observed over 50 h in mouse plasma. More than 50% of the conjugate released free TP during 50 h in cell culture medium. The results indicate that the parent TP is released from the conjugate via cleavage of a succinate linker in vitro, so that TP-CMCS can be considered as a prodrug.

### 3.2. In Vitro Cytotoxicity of TP-CMCS

We used the non-activated macrophage cells to obtain a profile of the cytotoxicity of TP-CMCS, reasoning that the cytotoxicity of TP-CMCS should be lower than that of free TP. Figure 2 shows the in vitro cytotoxicity of TP and TP-CMCS against RAW264.7 cell (macrophage cell). The half-inhibition concentrations (IC50) of TP and TP-CMCS are 41.47 ± 8.87 nmol/L vs. 221.86 ± 12.87 nmol/L at 24 h and 27.14 ± 10.21 nmol/L vs. 120.81 ± 21.23 nmol/L at 48 h, respectively (Table 1). The TP drug delivery system decreases the growth inhibition by about fivefold, which could be accounted for by the slow release of TP from the conjugate. However, the survival rates of TP-CMCS and TP (222 nmol/L of TP) treated groups were nearly the same at 48 h, which suggests that the TP were almost released from the TP-CMCS. These data suggest that TP-CMCS can reduce toxicity and other side effects in the inhibition of macrophage proliferation.

### 3.3. TP-CMCS Decreased TP-Induced Apoptosis

The effect of the TP drug delivery system on TP-induced apoptosis of macrophage cells was tested using a flow cytometric method, as shown in Figure 3. The in vitro experiments showed that, compared with the control group, the early apoptosis rates gradually increased from 5.69% and 40.06% to 42.26% and then decreased to 34.4%, and the late stage apoptosis rates increased from 11.74%, 39.26%, 47.31%, and 47.31% to 57.32% after treatment with 27.5, 55.5, 111, 222 nmol/L of triptolide, respectively, for 24 h (Figure 3A–D). This suggests that TP can induce cell apoptosis and causes a significantly increase with a concentration-dependent manner. After incubation with TP-CMCS (at equivalent TP concentrations) for 24 h, the apoptosis rates of macrophage cells were slightly increased from 4.55%, 2.69%, and 11.6% to 22.93% (at early apoptosis stage) and from 7.83%, 12.58%, and 14.87% to 33.49% (at late apoptosis stage) (Figure 3E–H). This indicates that the TP drug delivery system has protective effects on TP-induced macrophage apoptosis. The results demonstrated that the apoptotic cells were less in the TP-CMCS group than those in the TP group.

### 3.4. In Vivo Toxicity of TP-CMCS

In vivo toxicity of TP-CMCS was detected by using healthy Kunming mice. The survival rates of mice were 100% after administration with PBS (Figure 4A) and TP-CMCS at up to 2 mg TP equivalent/kg (Figure 4B–E), whereas all mice died after 4 days administration with 2 mg/kg TP (Figure 4E), and only 40% of the mice survived after 6 days administration with 1 mg/kg TP (Figure 4D). However, the survival rates were 87.5% and 75% on the second day when mice were injected with 0.25 and 0.5 mg/kg TP, respectively (Figure 4B,C). In the survival mice, all TP-CMCS groups demonstrated body weight nearly similar to that of the control group, while the TP groups showed obvious weight reduction (Figure 5A). At the end of the experiment, 0.5 and 1 mg/kg of TP administration caused a 34.4% and 38.3% loss of body weight (*p* < 0.0001), respectively, while TP-CMCS administration with equivalent TP concentration showed no adverse effect on body weight (Figure 5B). The studies indicated that TP-CMCS conjugate significantly protected mice from toxicity injury of TP.

### 3.5. Therapeutic Effect on CIA Model

CIA models were used to evaluate the potential efficacy of TP-CMCS conjugate in vivo. Figure 6, Figure 7 and Figure 8 showed the results of weight, foot volume, and arthritis deformation indexes of mice before and after treatment. Compared with the control group (normal mice, as above described), all CIA mice groups showed varying degrees of body weight loss due to the toxicity of TP or/and the systemic inflammation (Figure 6A). In particular, the body weight of the TP-treated mice did not increase but dropped noticeably during treatment. However, the body weight of the mice in all other TP-CMCS groups increased remarkably, but with no significant difference. At the end of the experiment, the TP-CMCS groups displayed body weight nearly similar to that of the control group, but the TP group showed obvious weight reduction (*p* < 0.01) (Figure 6B). The results further confirmed that TP-CMCS could effectively alleviate the toxicity injury of TP and the systemic inflammation, compared with the TP group.

Both the foot volume and arthritis deformation indexes were used to describe the arthritis symptoms of paws, shown in Figure 7 and Figure 8. During the first seven days after administration, the claws of all the arthritic mice were very similar in size, and subsequently, the foot volume of the treatment groups was between that of the model group and that of control group (Figure 7A). At the end of the experiment, the foot volume in the treatment groups was significantly lower than that of the model group, and the foot volume values in the TP group were almost the same as those in the control group (Figure 7B). In addition, the arthritis deformation indexes were used to describe the shape and joint swelling of the paws of CIA mice (Figure 8). Compared with the model group, the detumescence rate of paws in the treatment groups increased gradually after 7 days administration, and the change of the index in the TP group was more notable than in other groups (*p* < 0.05). The results were highly consistent with that of the foot volume curve, and the index in the TP group first reached the normal level after 20 days administration, which indicated that TP could more quickly alleviate the joint swelling of mice than TP-CMCS.

Blood serum analysis demonstrated that only TP significantly induced liver and kidney injury, whereby the level of AST, ALT, BUN, and Crea in the TP group was two to four times higher than that in the control group. And there were no obvious differences between 0.5 mg/kg TP-CMCS group and control group, which indicated that TP-CMCS is lower toxic than TP in CIA mice (Table 2).

The level of IL-1β, IL-6, and TNF-α in the serum was detected using ELISA (Figure 9). Compared with the control group, levels of the pro-inflammatory cytokines in the serum of the model group were significantly high (*p* < 0.05), suggesting that the arthritis model induced by collagen was successful. Levels of the pro-inflammatory cytokines in the serum of the 2 mg/kg TP-CMCS group and the TP group were lower than those of the model group, and there was no obvious difference in IL-6 and IL-1β between all treatment groups (*p* > 0.05), while there was significant difference in TNF-α (*p* < 0.01). The results from Figure 9 indicate that 1 mg/kg and 2 mg/kg of TP-CMCS and 0.5 mg/kg TP could effectively reduce the IL-1β, IL-6, and TNF-α level in the serum of CIA mice, and the in vivo anti-inflammatory effects of TP-CMCS were dose-dependent.

Histological examination revealed different degrees of erosion on the surface of the synovial membrane and bone tissue, compared with the control group (Figure 10B). After administration, the synovitis and bone erosion of mice in the TP-CMCS group and TP group were effectively relieved (Figure 10C–F), indicating that TP-CMCS (especially with 2 mg/kg) and 0.5 mg/kg TP could alleviate joint inflammation by down-regulating pro-inflammatory cytokine levels in CIA mice.

## 4. Conclusions

TP-CMCS was successfully prepared containing ~5 wt% TP and showing high water solubility (>5 mg/mL), which is suitable for TP drug delivery in rheumatoid arthritis treatment. The in vitro experiments showed that TP-CMCS could relieve TP-induced inhibition on RAW264.7 cells and apoptosis, respectively. Compared with the TP group, TP-CMCS could improve the survival rate of the mice (*p* < 0.05). TP-CMCS could effectively alleviate the swelling and deformation of CIA mice and down-regulate TNF-α, IL-1β, and IL-6 levels to inhibit the erosion of synovitis and bone tissue. Our results demonstrated that TP-CMCS is a promising drug delivery system for the treatment of RA, which enhances water solubility of free TP and reduces drug toxicity in vitro/vivo. However, further research is needed to explore its mechanism, toxicity, and other side effects, as well as the optimal dose for human beings.

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
