# Peer review of "Novel Carboxylated Chitosan-Based Triptolide Conjugate for the Treatment of Rheumatoid Arthritis"

_pharmaceutics, 2020, doi:10.3390/pharmaceutics12030202_

Round 1

Reviewer 1 Report

This manuscript describes the preparation of triptolide-carboxylated chitosan for treatment of rheumatoid arthritis. Conjugation of triptolide to carboxylated chitosan significantly reduced its cytotoxicity both in vitro and in vivo. Intravenous injection of triptolide-carboxylated chitosan relieved inflammatory responses at paws of CIA model mice. Therefore, chemical conjugation of triptolide to polysaccharide would provide good strategy for effective biotherapeutics on inflammatory-relating diseases. However, current manuscript still contains many points to be addressed as follows:

-Lines 25-31 repeats almost same contents about rheumatoid arthritis. The author should be corrected these sentences.

-Line 44: “Generally, it is more effective in the treating RA” than for what?

-Introduction: The author should describe the rationale of the use of carboxylated chitosan as a base polymer for conjugation more clearly. Other carboxylated polysaccharides such as carboxymethyl cellulose is also suitable for the author’s purpose.

-Figure 8: Maximum arthritis score should be 16 (not 4) by counting all four paws for each mouse. In addition, statistical analysis should be performed to discuss the difference between samples.

-Detection of anti-collagen type II antibody in the blood is crucial to discuss therapeutic effects.

Author Response

Point 1: Lines 25-31 repeats almost same contents about rheumatoid arthritis. The author should be corrected these sentences.

Response 1: These sentences are repetitious with some mistakes and deleted from the original.

Point 2: Line 44: “Generally, it is more effective in the treating RA” than for what?

Response 2: We rewrote this sentence on the original: Generally, it is more effective in treatment of RA than in that of systemic lupus.

Point 3: Introduction: The author should describe the rationale of the use of carboxylated chitosan as a base polymer for conjugation more clearly. Other carboxylated polysaccharides such as carboxymethyl cellulose is also suitable for the author’s purpose.

Response 3: We have made some additions to the rationale of the use of carboxylated chitosan as follows. Here, we choose just the carboxymethyl chitosan as the drug delivery carrier.

“To address these issues, an immunosuppressant conjugate should be developed for RA treatment. Currently, chitosan usually is widely explored as a drug delivery carrier. However, its applications are limited by its poor solubility in water. Carboxylmethyl chitosan (CMCS), a chemical modification version of chitosan, was used to form ……”

Point 4: Figure 8: Maximum arthritis score should be 16 (not 4) by counting all four paws for each mouse. In addition, statistical analysis should be performed to discuss the difference between samples.

Response 4: We have replaced the original Figure 8 as required by the reviewer. The maximum arthritis score for each mouse was not more than 16 points in the new Figure 8, and the results of statistical analysis were shown below the chart heading.

Point 5: Detection of anti-collagen type II antibody in the blood is crucial to discuss therapeutic effects.

Response 5: We appreciate your constructive suggestions. In our work, we focus on the performance of TP-CMCS, such as increasing the water solubility, reducing the cytotoxicity of TP and increasing or maintaining its efficacy. Normally, the serum concentrations of IgG, IgG1, and IgG2a can reflect directly the therapeutic effects of TP-CMCS on collagen-induced arthritis. So detection of anti-collagen type II antibody in the blood is very important to discuss therapeutic effects. Regrettably, we only applied some indicators, such as body weight, foot volume and arthritis deformation indexes as well as biochemical parameters of blood serum, histological examination, to evaluate the therapeutic effects and the drug toxicity reduction, which does not disturb the conclusion of the therapeutic effect of TP. There are many similar studies to prove this point, as following literatures. In the follow-up study of TP-CMCS, we will perfect the detection index of IgG, IgG1, and IgG2a to discuss therapeutic effects.

Literatures:

Journal of Controlled Release 2020, 319:87-103;

Journal of Controlled Release 2015, 216:140-148;

International Journal of Nanomedicine 2019:14 8561–8572;

International Journal of Pharmaceutics 2019, 554:235-244.

Reviewer 2 Report

The current manuscript provides an interesting account of Chitosan-Based Triptolide Conjugate and tested the same for RA intervention. I recommend some revisions for the manuscript as follows:

  1. Section 2.10: The sentence "After the treatment, the blood samples were collected from eyes; the arthritic paws were collected in saline and stored at −80 °C." is not clear and should be revised.
  2. Figure 1: The release study should have been conducted for longer (atleast 48 hours) to atleast match the 2 day dosing.
  3. The authors are asserting the conjugate as a prodrug and further stated in line 200 that "The results indicate that the parent TP is released from the conjugate via cleavage of a succinate linker in vitro, so that TP-CMCS can be considered as a prodrug." The authors provided no results suggesting the involvement of succinate linker?
  4. Figure 1 and Figure 8: It will be interesting to see the IVIVC in this study.

Author Response

Point 1: Section 2.10: The sentence "After the treatment, the blood samples were collected from eyes; the arthritic paws were collected in saline and stored at −80 °C." is not clear and should be revised.

Response 1: We revised this sentence on the original: After the treatment, blood samples were collected from the eyes; arthritic paws were collected for histological examination.

Point 2: Figure 1: The release study should have been conducted for longer (at least 48 hours) to at least match the 2 day dosing.

Response 2: We extended the stability-test time and modified the graph in Figure 1.

Point 3: The authors are asserting the conjugate as a prodrug and further stated in line 200 that "The results indicate that the parent TP is released from the conjugate via cleavage of a succinate linker in vitro, so that TP-CMCS can be considered as a prodrug." The authors provided no results suggesting the involvement of succinate linker?

Response 3: When HPLC was used to detect the stability of the conjugate, we found that the retention time or peak of TP cleaved from the conjugate was consistent with that of the original drug; and the literature also proved this viewpoint from another aspect.

“Journal of Controlled Release 2009, 140:79-85”.

Point 4: Figure 1 and Figure 8: It will be interesting to see the IVIVC in this study.

Response 4: The sustained release effect of the conjugates (Figure 1) can reduce the concentration and cytotoxicity of free TP in blood circulation, and also reduce the concentration of TP for treatment of RA, but the treatment time of TP for RA was prolonged. Therefore, Figure 8 showed that the detumescence ability of TP was generally higher than that of the conjugates.

Round 2

Reviewer 1 Report

The author revised the manuscript properly according to the reviewer’s comments. Current manuscript is acceptable for publication.

Reviewer 2 Report

The authors have successfully addressed the comments raised by the reviewer.